# Temperature-dependent rearrangement of gas molecules in ultramicroporous materials for tunable adsorption of $CO_2$ and $C_2H_2$

Zhaoqiang Zhang [1,8], Yinlin Chen [2,8], Kungang Chai[3], Chengjun Kang[1], Shing Bo Peh [1], He Li[1], Junyu Ren[1], Xiansong Shi[1], Xue Han[4], Catherine Dejoie [5], Sarah J. Day[6], Sihai Yang [2,7] ✉ & Dan Zhao [1] ✉

The interactions between adsorbed gas molecules within porous metal-organic frameworks are crucial to gas selectivity but remain poorly explored. Here, we report the modulation of packing geometries of $CO_2$ and $C_2H_2$ clusters within the ultramicroporous CUK-1 material as a function of temperature. In-situ synchrotron X-ray diffraction reveals a unique temperature-dependent reversal of $CO_2$ and $C_2H_2$ adsorption affinities on CUK-1, which is validated by gas sorption and dynamic breakthrough experiments, affording high-purity $C_2H_2$ (99.95%) from the equimolar mixture of $C_2H_2/CO_2$ via a one-step purification process. At low temperatures (<253 K), CUK-1 preferentially adsorbs $CO_2$ with both high selectivity (>10) and capacity (170 $cm^3$ $g^{-1}$) owing to the formation of $CO_2$ tetramers that simultaneously maximize the guest-guest and host-guest interactions. At room temperature, conventionally selective adsorption of $C_2H_2$ is observed. The selectivity reversal, structural robustness, and facile regeneration of CUK-1 suggest its potential for producing high-purity $C_2H_2$ by temperature-swing sorption.

Host-guest chemistry is fundamental to the selectivity of many molecular recognition systems[1–5]. The optimization of cooperative interactions, such as electrostatic interactions and hydrogen bonding, plays a crucial role in the design of efficient molecular recognition systems, particularly in porous materials. These cooperative interactions are essential for achieving high performance in gas adsorption, sensing, and catalysis applications[1,3,6–17]. On the other hand, guest-guest interactions or the formation of guest clusters also play an important role in molecular recognition. However, the direct observation and control of guest-guest interactions within confined nanovoids of porous materials is highly challenging and remains poorly explored[18–20]. Screening new host-guest and guest-guest interactions can promote the design of new functional porous materials[1,21].

Ultramicroporous metal-organic frameworks (MOFs), featuring highly inerratic porosity, tunable pore chemistry, and designable structures, provide a unique platform to explore host-guest interactions[11,22–27]. In particular, the modular nature and reticular structure endow ultramicroporous MOFs with the possibility to precisely control the host-guest and guest-guest interactions within the pores[28,29]. Great advances in host-guest chemistry have been achieved in ultramicroporous MOFs with tailor-made properties for gas adsorption and separation, owing to the confinement effect from the strong host-guest interactions[30–32]. Currently, the major interest in gas

[1]Department of Chemical and Biomolecular Engineering, National University of Singapore, 117585 Singapore, Singapore. [2]Department of Chemistry, The University of Manchester, Manchester M13 9PL, UK. [3]School of Chemistry and Chemical Engineering, Guangxi University, Nanning 530004, China. [4]College of Chemistry, Beijing Normal University, Beijing 100875, China. [5]The European Synchrotron Radiation Facility, 71 Avenue des Martyrs, CS40220 Cedex 9, 38043 Grenoble, France. [6]Diamond Light Source, Harwell Science Campus, Oxfordshire OX11 0DE, UK. [7]College of Chemistry and Molecular Engineering, Beijing National Laboratory for Molecular Sciences, Peking University, Beijing 100871, China. [8]These authors contributed equally: Zhaoqiang Zhang, Yinlin Chen. ✉e-mail: sihai.yang@manchester.ac.uk; chezhao@nus.edu.sg

adsorption and separation using ultramicroporous MOFs is focused on enhancing recognition selectivity by tuning the host-guest interactions[19,29,33–36]. This is pronounced for selective adsorption of acetylene ($C_2H_2$) from carbon dioxide ($CO_2$), as $C_2H_2$ is one of the most important industrial precursors, and the $CO_2$ contaminant would be coproduced during the production of $C_2H_2$ via partial combustion of natural gas[34–36]. However, the understanding of the impact of guest-guest interactions or guest clusters on selectivity remains lacking due to the difficulties in the direct observation of such dynamic and weak interactions.

Herein, we report the modulation of geometries of guest-clusters as a function of temperature (Fig. 1) for the normal and inverse selectively and separation of $CO_2$ and $C_2H_2$ within the robust ultra-microporous M-CUK-1 (M = Co, Ni, and Mg) materials. The guest-guest interactions and binding domains within CUK-1 with different metal nodes have been observed by in-situ synchrotron X-ray diffractions and molecular simulations. The efficient packing of well-organized $CO_2$ clusters with T-shaped dimers gives rise to notably higher crystallographic occupancy and capacity of $CO_2$ (106 vs. 86 cm$^3$ g$^{-1}$ of $C_2H_2$ in Co-CUK-1 at 298 K), while the stronger host-guest interactions between $C_2H_2$ and CUK-1 at room temperature lead CUK-1 to preferentially adsorb $C_2H_2$ over $CO_2$ (Fig. 1). Notably, a much larger increment of $CO_2$ capacities at low temperatures was observed compared with those of $C_2H_2$, which is resulted from the highly efficient packing of $CO_2$ clusters with tetramers and the significantly stronger host-guest interactions between $CO_2$ and CUK-1. This finally leads to much higher $CO_2$ capacities (170 vs. 119 cm$^3$ g$^{-1}$ of $C_2H_2$ at 233 K) and clear sorption inversion of $CO_2$ over $C_2H_2$. Such an inverse $CO_2/C_2H_2$ adsorption behavior is more desirable for industrial production of $C_2H_2$ via a one-step $CO_2$ adsorption process but is rarely investigated[18,29,37–40]. The temperature-dependent reversal of sorption behavior for $CO_2$ and $C_2H_2$ is demonstrated by gas sorption isotherms and dynamic breakthrough experiments at various temperatures. High-purity $C_2H_2$

(99.995%) can be directly obtained in a one-step process, and the low energy input for the regeneration suggests that CUK-1 is a promising adsorbent for $C_2H_2$ production via the temperature-swing adsorption (TSA) process.

## Results

### Materials and characterization

M-CUK-1 (M = Co, Ni, and Mg) were hydrothermally synthesized by reacting 2,4-pyridinedicarboxylic acid (2,4-H$_2$pdc) and M$^{2+}$-containing salts (M = Co, Ni, and Mg) with KOH in water at 210 °C for 24 h[41–43]. The CUK-1 materials are isostructural. The edge- and vertex-sharing M$_3$($\mu_3$-OH)$_2$ chains serve as undulating pillars connecting the 2,4-pdc ligands in an orthogonal fashion, forming a 'wine-rack' topology with one-dimensional diamond-shaped and corrugated channels (Supplementary Fig. 1)[41–43]. All three CUK-1 materials show excellent chemical and structural stability, and are entirely stable upon air exposure for two years (Supplementary Figs. 2–5). Desolvated CUK-1 exhibits an ultra-microporous structure, as evidenced by the negligible N$_2$ uptakes and typical type-I $CO_2$ isotherms at 77 K and 196 K, respectively (Supplementary Figs. 6–9). The calculated Brunauer–Emmett–Teller (BET) surface areas are 500-600 m$^2$ g$^{-1}$ based on the $CO_2$ isotherms. Upon desolvation, the exposed $\mu_3$-OH groups reside orderly in the channels (8.1 × 10.6 Å$^2$, Supplementary Fig. 1), acting as potential binding sites to guest molecules through electrostatic interactions[41–43]. This is highly desirable for the adsorption and separation of hydrocarbons.

### Analysis of gas adsorption isotherms and selectivity

Adsorption isotherms of $CO_2$ and $C_2H_2$ on desolvated M-CUK-1 (M = Co, Ni, and Mg) at 298 K indicate the preferential adsorption of $C_2H_2$ at low pressure but higher saturation capacity of $CO_2$ upon increasing the pressure (Fig. 2a–c). This behavior results in the intersection of the two isotherms at moderate pressure. Another ultramicroporous compound, SIFSIX-3-Ni, exhibits similar isotherm crossing but with a stronger affinity to $CO_2$ at low pressures[29]. After the intersection of $CO_2$ and $C_2H_2$ isotherms at 0.42 bar on Co-CUK-1, the $CO_2$ isotherm is above that of $C_2H_2$, and $CO_2$ uptake at 1 bar can reach 106 cm$^3$ g$^{-1}$, much higher than that of $C_2H_2$ (86 cm$^3$ g$^{-1}$, Fig. 2a). To the best of our knowledge, such an intersection between $CO_2$ and $C_2H_2$ isotherms is rarely observed on porous materials[29]. Similarly, Ni- and Mg-CUK-1 show stronger sorption affinities to $C_2H_2$ in the low-pressure range, and the $CO_2$ and $C_2H_2$ isotherms also intersect but with relatively higher intersecting pressures of 0.6 and 0.95 bar on Ni- and Mg-CUK-1, respectively (Fig. 2b, c).

Considering that the inversed $CO_2/C_2H_2$ selectivity is more desirable for industrial $C_2H_2$ production, the respective guest loadings were measured at progressively lower temperatures to decrease the intersecting pressures. Upon reducing the temperature, there are significant enhancements for $CO_2$ uptakes but only a slight increase in $C_2H_2$ uptakes, finally leading to much higher $CO_2$ uptakes, even at very low pressures (Fig. 2a–c and Supplementary Figs. 7–9). Specifically, the $CO_2$ uptakes at 233 K are 170, 142, and 144 cm$^3$ g$^{-1}$ on Co-, Ni-, and Mg-CUK-1 (ca. 4.15, 3.43, and 3.50 $CO_2$ molecules per cell, respectively), which notably exceed those of $C_2H_2$ (119, 97, and 89 cm$^3$ g$^{-1}$, respectively; ca. 2.89, 2.29, and 2.32 $C_2H_2$ molecules per cell on Co-, Ni-, and Mg-CUK-1, respectively). The densities of adsorbed $CO_2$ molecules (based on the structural pore volume) in Co-, Ni-, and Mg-CUK-1 at 233 K were estimated to be 1.40, 1.27, and 1.25 g cm$^{-3}$, respectively. Notably, these densities are higher than that of liquid $CO_2$ (1.1 g cm$^{-3}$) but lower than that of dry ice (1.55 to 1.7 g cm$^{-3}$)[44], indicating the highly efficient packing of $CO_2$ in the pores. However, the densities of adsorbed $C_2H_2$ molecules were recorded as only 0.56, 0.50, and 0.45 g cm$^{-3}$ in Co-, Ni-, and Mg-CUK-1, respectively, lower than that of liquid $C_2H_2$ (0.69 g cm$^{-3}$)[45]. At 253 K and 233 K, there is a clear inversion in the adsorption selectivity from $C_2H_2$ to $CO_2$ on the CUK-1 materials. The uptake gap between $CO_2$ and $C_2H_2$ at 0.5 bar on

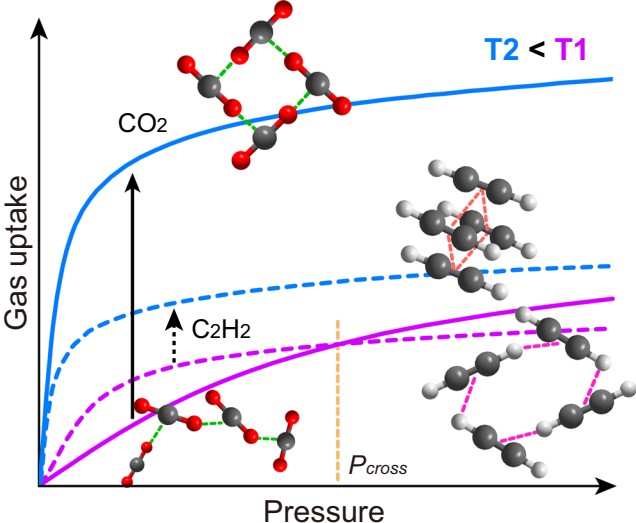

**Fig. 1 | Illustration of the temperature-dependent packing geometries of guest clusters for the normal and inverse adsorption behavior.** At a high temperature (T1, purple), the strong host-guest and guest-guest interactions result in preferential adsorption of $C_2H_2$, but the efficient packing of molecular chains formed by $CO_2$ molecules through strong guest-guest interactions leads to the higher uptake at the high-pressure range (>$P_{cross}$). After decreasing the temperature to T2 (blue), the $CO_2$ clusters with T-shaped dimers exhibit higher occupancy of the pore channels than that of $C_2H_2$ clusters packed together via π···π interactions, coupled with the strong host-guest interactions, leading to the inverse $CO_2$ preferential sorption.

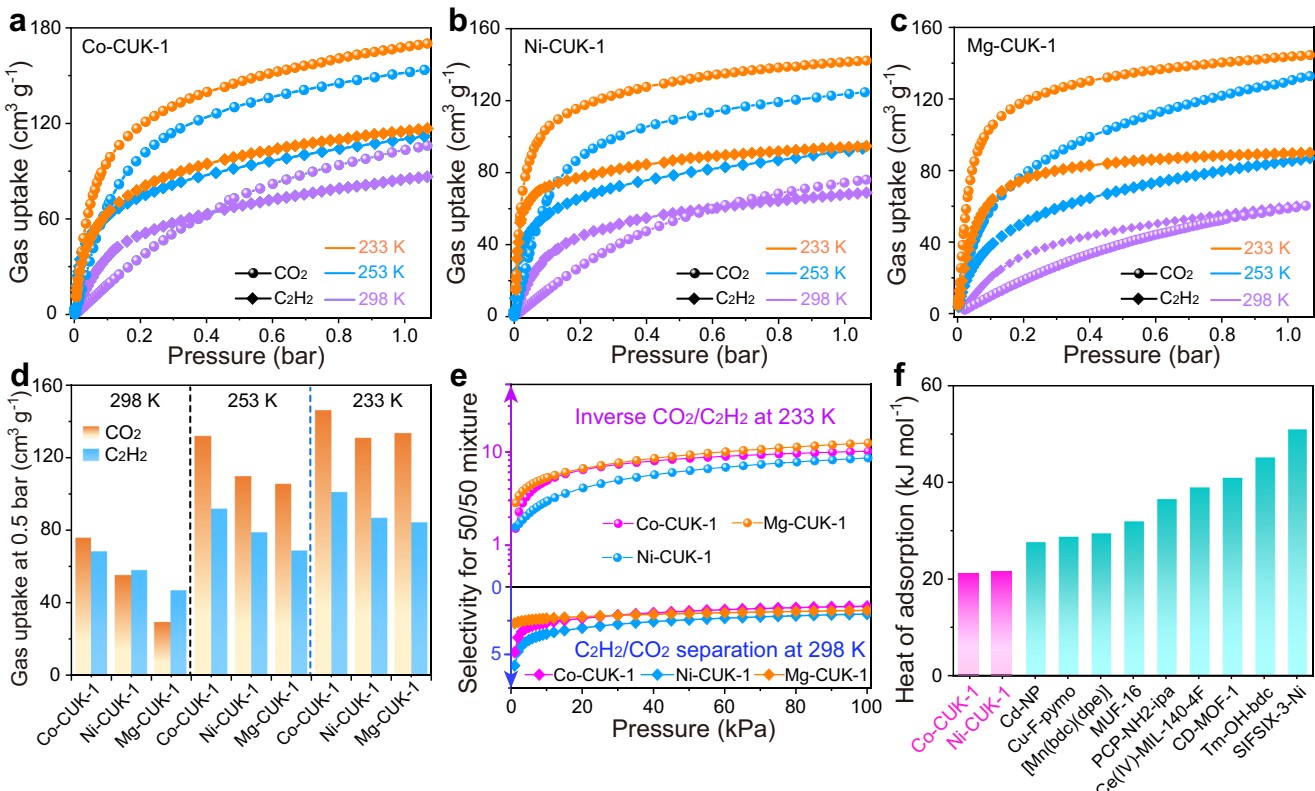

**Fig. 2 | CO2 and C2H2 adsorption and separation performance on CUK-1 materials.** The $CO_2$ and $C_2H_2$ adsorption isotherms on desolvated Co-CUK-1 (**a**), Ni-CUK-1 (**b**), and Mg-CUK-1 (**c**) at 298 (purple), 253 (blue), and 233 K (orange). **d** The comparison of $CO_2$ and $C_2H_2$ uptakes at 0.5 bar on CUK-1 materials at different temperatures. **e** The inverse $CO_2/C_2H_2$ (1/1) (top) and normal $C_2H_2/CO_2$ (1/1) (bottom) selectivities at 233 and 298 K, respectively, on CUK-1 materials. **f** Comparison of the zero-coverage heat of adsorption of CUK-1 materials for $CO_2$ with those of other materials for inverse $CO_2/C_2H_2$ separation.

Co-CUK-1 can reach 40 and 45 cm³ g⁻¹ at 253 and 233 K (Fig. 2d), respectively.

State-of-the-art $C_2H_2/CO_2$ separation is mainly realized by cryogenic distillation and solvent absorption with high energy penalty. The adsorptive separation using $CO_2$-selective other than $C_2H_2$-selective materials is preferable in the industry for producing pure $C_2H_2$ via one-step sorption procedures. To see whether such a temperature-induced adsorption inversion behavior can be used for inverse $CO_2/C_2H_2$ separation, we evaluated CUK-1 materials for separating the equimolar mixture of $CO_2/C_2H_2$ by analyzing the single-component isotherms via ideal adsorbed solution theory (IAST). At 298 K, CUK-1 only shows a moderate $C_2H_2/CO_2$ selectivity of ca. 2. However, at 233 K, CUK-1 exhibits the inversed $CO_2/C_2H_2$ selectivity of 9.5, 8.4, and 12.1 for Co-, Ni-, and Mg-CUK-1, respectively (Fig. 2e). The inversed selectivities are comparable with those of the state-of-the-art materials for inversed $CO_2/C_2H_2$ separation, such as [Mn(bdc)(bpe)] (9)[46], Ce(IV)-MIL-140-4F (9.6)[37], PCP-NH2-ipa (6.4)[35], and SIFSIX-3-Ni (7.5)[29], but lower than the benchmark material Cu-F-pymo (> 10⁵)[38]. Furthermore, the isosteric heats of adsorption ($\Delta H_{ads}$) of $CO_2$ on Co- and Ni-CUK-1 were calculated to be 20.8 and 21.7 kJ mol⁻¹, respectively (Supplementary Fig. 10), much lower than that of other materials (Fig. 2f), such as PCP-NH2-ipa (26.8 kJ mol⁻¹)[35], [Mn(bdc)(bpe)] (29.5 kJ mol⁻¹)[46], MUF-16 (32 kJ mol⁻¹)[39], and $Tm_2$(OH-bdc) (45 kJ mol⁻¹)[40].

**Guest configurations determined by in-situ synchrotron X-ray powder diffraction**

In-situ synchrotron X-ray powder diffraction data on $CO_2$- and $C_2H_2$-loaded CUK-1 materials were collected as a function of temperature (Supplementary Figs. 11–12). Full refinements of the data indicate two binding sites in the asymmetric unit: site I is close to the $\mu_3$-OH group,

and site II locates near the pore surface (Figs. 3–4 and Supplementary Figs. 13–18). At 298 K, the total crystallographic occupancy of $C_2H_2$ molecules (2.05 per cell) in Co-CUK-1 is in excellent agreement with that obtained from the isotherm (2.07 $C_2H_2$ per cell). $C_2H_2$ molecules at site I locate almost perpendicular to $\mu_3$-OH groups, forming O-H···$\pi_{C2H2}$ H-bonds (2.92 Å, dotted green lines), supplemented by additional interactions via C-H$_{C2H2}$···O$_{ligand}$ H-bonding (dotted green lines, 2.67-2.71 Å, Fig. 3a and Supplementary Fig. 13a). $C_2H_2$ molecules at site II sit close to the aromatic rings on the pore surface and form weak interactions with the framework through multiple C-H$_{C2H2}$···O$_{ligand}$ H-bonding (3.37–3.82 Å) and $\pi_{C2H2}$···H$_{ligand}$ (3.44–3.67 Å) interactions. Moreover, at high loading, the neighboring $C_2H_2$ molecules synergistically interact with each other through multiple H$_{C2H2}$···$\pi_{C2H2}$ interactions (dotted pink lines, 2.33–2.95 Å), forming the tetramer-clusters of $C_2H_2$ (Fig. 3b).

In contrast, $CO_2$ molecules show different geometries of packing (Fig. 3c and Supplementary Fig. 14a). $CO_2$ molecules at site I exhibit an end-on interaction to $\mu_3$-OH group via hydrogen bonds (dotted lime lines, 2.42 Å of O-H···O$_{CO2}$), but no interactions between $CO_2$ and the ligand of CUK-1 were observed. $CO_2$ molecules at site II interact with the pore surface via weak O···H$_{ligand}$ interactions (3.88–3.91 Å, Supplementary Fig. 14a). Thus, adsorbed $CO_2$ molecules at both sites show much weaker interactions compared with $C_2H_2$, entirely consistent with the adsorption results at room temperature. However, at high loading, two one-dimensional chains of $CO_2$ (dotted azure lines) running along the channel were formed via strong guest-guest interactions (2.93 and 2.74 Å, Fig. 3d). These chains are stabilized by multiple weak intermolecular dipole interactions between monomer-to-dimer and dimer-to-dimer of $CO_2$. Furthermore, two chains interact with each other via multiple synergistic host-host interactions (dotted green

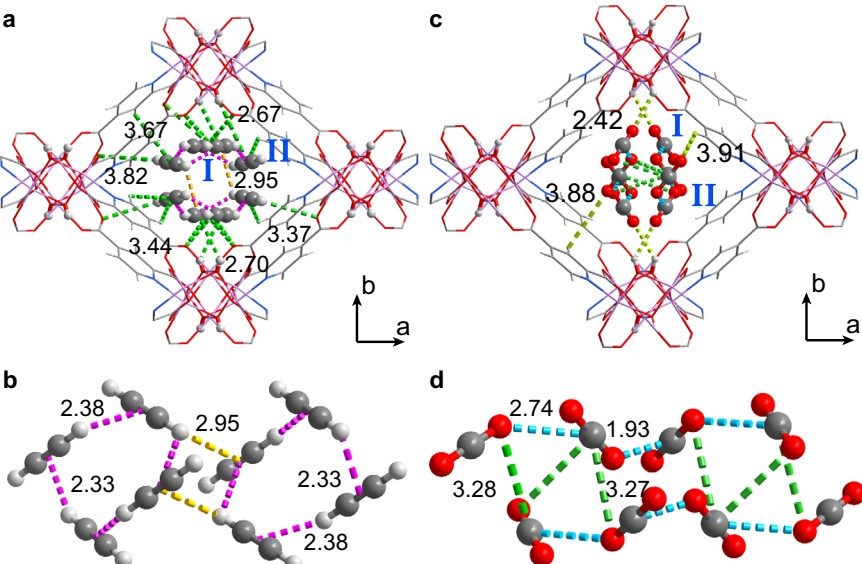

**Fig. 3 | Configurations of adsorbed C2H2 and CO2 molecules within Co-CUK-1 from refinements of in-situ synchrotron X-ray powder diffraction data at 298 K.** Views of host-guest interactions of $C_2H_2$ (**a**) and $CO_2$ (**c**) in Co-CUK-1. Packing geometries of $C_2H_2$ (**b**) and $CO_2$ (**d**) clusters. Color code: C, gray; H, gray-25%; O, red; N, blue; Co, pink.

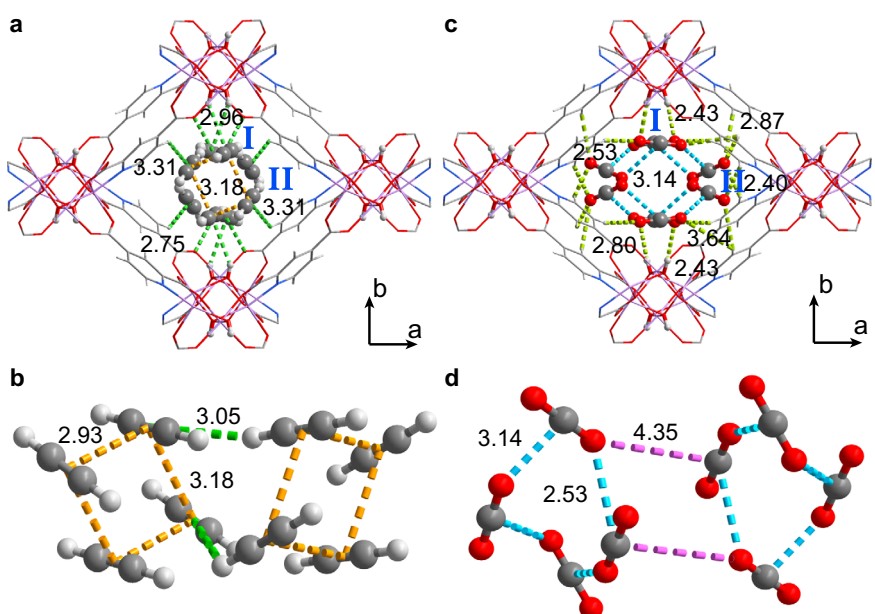

**Fig. 4 | Configurations of adsorbed C2H2 and CO2 molecules within Co-CUK-1 from refinements of in-situ synchrotron X-ray powder diffractions at 233 K.** Views of host-guest interactions between $C_2H_2$ (**a**) and $CO_2$ (**c**) in Co-CUK-1. Packing geometries of $C_2H_2$ (**b**) and $CO_2$ (**d**) clusters. Color code: C, gray; H, gray-25%; O, red; N, blue; Co, pink.

lines, 3.27–3.28 Å). Notably, the neighboring $CO_2$ molecules exhibit a head-to-center (C = O···C) geometry (Supplementary Fig. 14a), thus leading to the efficient packing of $CO_2$ in the pore channels. Similar binding sites of $CO_2$ in Ni-CUK-1 were also observed (Supplementary Fig. 18). Compared with $C_2H_2$ clusters, the efficient packing of $CO_2$ molecules near the center of pore channels via strong guest-guest interactions but with less host-guest interactions is the main reason for the high adsorption of $CO_2$ in CUK-1 at high pressures.

At 233 K, remarkable changes in the packing geometry of $CO_2$ and $C_2H_2$ were observed (Fig. 4), and the crystallographic occupancy of $C_2H_2$ molecules increased to 2.77 per cell. Meanwhile, the $CO_2$ occupancy increased to 3.47 per cell (vs. 2.07 at 298 K), indicating the high

capacity of Co-CUK-1 for $CO_2$ at 233 K compared with that for $C_2H_2$. This is entirely consistent with the isotherms. $C_2H_2$ molecules at site I interact with bridging $\mu_3$-OH groups via $\pi_{C2H2}$···H-O H-bond (2.96 Å) that is supplemented by weak C-H···$O_{ligand}$ H-bonding (dotted green lines, 2.75 Å, Fig. 4a and Supplementary Fig. 13b). $CO_2$ molecules at site I are stabilized by C-$O_{CO2}$···$H_{\mu3\text{-}OH}$ H-bonding (2.43 Å) and C-$O_{CO2}$···$H_{ligand}$ interactions (2.80 and 3.64 Å, Fig. 4c and Supplementary Fig. 14b). $C_2H_2$ molecules at site II reside near the center of pore channels with fewer host-guest interactions ($\pi$···$H_{ligand}$ 3.31 Å), similar to that of $CO_2$ in the channels at 298 K. However, $CO_2$ molecules are located in the corner of the pore channels and stabilized by multiple weak host-guest interactions (C-$O_{CO2}$···$H_{ligand}$, 2.40–2.87 Å), thus

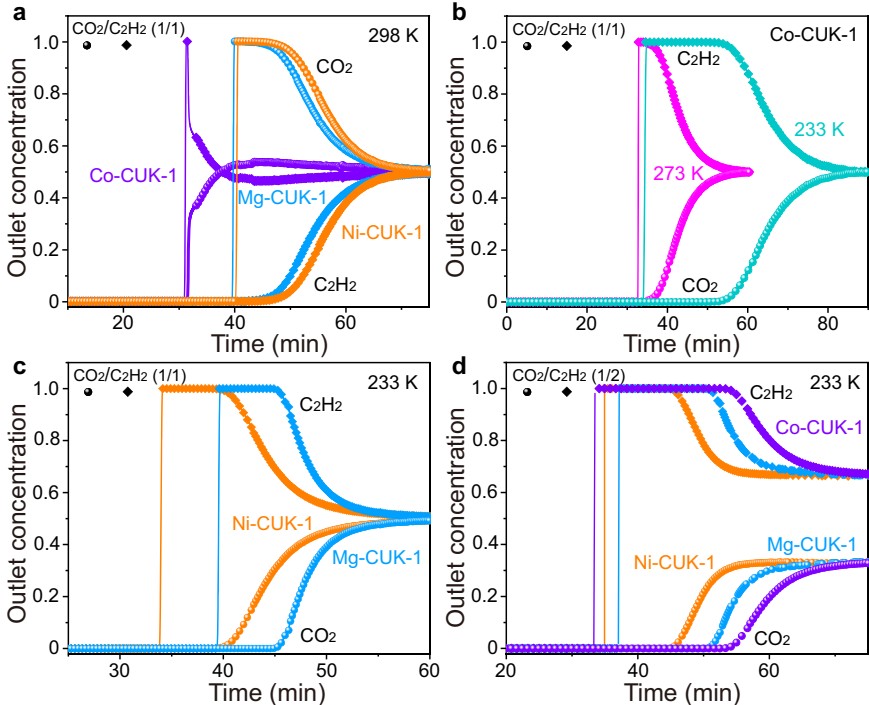

**Fig. 5 | Breakthrough curves of CUK-1 materials for CO2/C2H2 mixtures.**
**a** Breakthrough curves of $CO_2/C_2H_2$ (1/1) mixture on CUK-1 materials at 298 K with a flow rate of 2.0 mL min$^{-1}$. **b** Breakthrough curves of $CO_2/C_2H_2$ (1/1) mixture on Co-CUK-1 at 273 and 233 K with flow rates of 2.0 and 3.0 mL min$^{-1}$, respectively. **c** Breakthrough curves of $CO_2/C_2H_2$ (1/1) mixture on Ni-CUK-1 and Mg-CUK-1 at 233 K with a flow rate of 3.0 mL min$^{-1}$. **d** Breakthrough curves of $CO_2/C_2H_2$ (1/2) mixture on CUK-1 materials at 233 K with a flow rate of 3.0 mL min$^{-1}$.

leading to the strong binding affinities of host framework for $CO_2$. Meanwhile, in Co-CUK-1, $C_2H_2$ clusters are formed with $C_2H_2$ molecules via $\pi_{C2H2}\cdots\pi_{C2H2}$ interactions (dotted orange lines, 2.95 and 3.18 Å, Fig. 4b). The neighboring clusters synergistically interact with each other via weak C-H···π H-bonding (3.05 and 3.17 Å), leading to the efficient packing of $C_2H_2$ molecules. By contrast, the isolated $CO_2$ clusters are formed with four $CO_2$ molecules by closely interacting with each other (distances of 2.53 and 3.14 Å) with the head-to-center configurations, forming the quasi-T-shaped geometry (C = O···C, dotted azure lines, Fig. 4d). This is similar to that in dry ice, indicating the highly efficient packing of $CO_2$ molecules (thus packing densities) in Co-CUK-1. Similar $CO_2$ clusters with quasi-T-shaped dimers were also observed in Ni-CUK-1 at 233 K (Supplementary Fig. 16d). Thus, the notably stronger guest-guest interactions between adsorbed $CO_2$ molecules at low temperature promote the unusually selective adsorption of $CO_2$ over $C_2H_2$ at 233 K. Importantly, to the best of our knowledge, such guest-guest packing geometries and host-guest interactions at different temperatures have not been observed in porous materials yet.

To quantitatively compare the binding affinities of CUK-1 to $CO_2$ and $C_2H_2$ at different temperatures, the static binding energies ($\Delta E$) were further estimated by first-principles density functional theory (DFT) calculations (Supplementary Tables 2 and 3). The results show that after decreasing the temperature from 298 to 233 K, there are significant increases of $\Delta E$ at site I for $CO_2$. Especially, $\Delta E$ for $CO_2$ at site I on Co-CUK-1 is 43.5 kJ mol$^{-1}$, much higher than that for $C_2H_2$ (33.3 kJ mol$^{-1}$), directly validating the preferential adsorption of $CO_2$ over $C_2H_2$. There is a subtle difference in host-guest interactions when varying the metal nodes in CUK-1 (Supplementary Tables 2 and 3), and this has little influence on the formation of guest clusters and the tunable $CO_2$ and $C_2H_2$ sorption behavior. Thus, the guest-guest interactions and/or the arrangement of guest clusters play the dominant role in the inverse $CO_2$ sorption of CUK-1 materials.

## Dynamic breakthrough tests
Dynamic breakthrough experiments on CUK-1 materials using mixtures of $CO_2/C_2H_2$ were conducted (Fig. 5). For the equimolar $CO_2/C_2H_2$ mixture at 298 K, Ni- and Mg-CUK-1 show typical $C_2H_2$-preferential sorption over $CO_2$ with a clear separation of $C_2H_2$ and $CO_2$, but Co-CUK-1 shows very poor separation (Fig. 5a and Supplementary Fig. 19). These are consistent with the isotherm results at 298 K. At 273 K, a clear inversed $CO_2/C_2H_2$ separation was observed on Co-CUK-1 (Fig. 5b and Supplementary Fig. 20), and there is an obvious deterioration in $C_2H_2/CO_2$ separation performance on Ni- and Mg-CUK-1 (Supplementary Fig. 21). At 233 K, an evident inversed $CO_2/C_2H_2$ separation was observed on CUK-1 materials, and all materials exhibit highly selective adsorption of $CO_2$ over $C_2H_2$ (Fig. 5b, c). The dynamic $CO_2$ uptake capacities at 233 K were calculated to be 140, 110, and 122 cm$^3$ g$^{-1}$ on Co-, Ni-, and Mg-CUK-1, respectively, much higher than those of $C_2H_2$ (62, 67, and 78 cm$^3$ g$^{-1}$, respectively). To mimic the industrial processes for $C_2H_2$ production, we further studied a gas mixture of $CO_2/C_2H_2$ (1/2). A complete inversed $CO_2/C_2H_2$ separation was realized with $CO_2$ retained in the fixed bed for a longer duration (Fig. 5d). Significantly, the productivity of pure $C_2H_2$ (99.995%) can reach 62 and 41.7 L kg$^{-1}$ on Co-CUK-1 and Mg-CUK-1, respectively, much higher than that on MUF-16 (27 L kg$^{-1}$)[39] and Cu-F-pymo (3.7 L kg$^{-1}$)[38]. It is worth noting that these materials show excellent cycling separation performance and can be easily regenerated by purging helium (He) at 298 K (Supplementary Figs. 22–23). The notably high $CO_2$ uptake and facile regeneration of CUK-1 are particularly desirable for practical applications to reduce the energy footprint compared with state-of-the-art cryogenic distillations.

## Discussion
In summary, we report the direct observation of packing geometry rearrangement of gas clusters as a function of temperature to control the adsorption selectivity of $CO_2$ and $C_2H_2$ on ultramicroporous MOFs.

The strong host-guest interactions of CUK-1 for $C_2H_2$ at ambient conditions led to the preferential adsorption of $C_2H_2$. However, the efficient packing of $CO_2$ molecules via strong guest-guest interactions forms $CO_2$ clusters, leading to higher $CO_2$ capacity. Impressively, after decreasing temperature, the host-guest interactions between $CO_2$ and host framework became stronger than that for $C_2H_2$. Furthermore, the highly ordered arrangement of $CO_2$ clusters with the T-shaped dimers endows CUK-1 with remarkably higher capacities for $CO_2$ over those for $C_2H_2$. Such host-guest interactions, guest-guest interactions, and gas cluster formation were elucidated by in-situ synchrotron X-ray powder diffraction and molecular simulations. This idiosyncratic inversion of the adsorption behavior of $C_2H_2$ and $CO_2$ was verified by dynamic breakthrough experiments with high-purity $C_2H_2$ (99.995%) obtained in a one-step process. Furthermore, the fixed-bed packed with CUK-1 can be easily regenerated at room temperature by purging an inert gas, indicating that the TSA process using CUK-1 materials is highly efficient for $C_2H_2$ production.

## Methods

### Gas adsorption and separation experiments
$CO_2$ and $C_2H_2$ sorption isotherms were collected at different temperatures on a Micromeritics ASAP 2020 instrument equipped with commercial software for data calculation and analysis. The test temperatures were controlled by soaking the sample cell in a circulating water bath (298 K), ice/methanol mixture (233–273 K), dry ice/acetone mixtures (196 K), or liquid $N_2$ (77 K). Before measurement, the sample (80–100 mg) was degassed at 423 K for 24 h. The breakthrough experiments were performed in a stainless-steel fixed bed (4.6 mm inner diameter × 50 mm length) packed with ~0.6 g of CUK-1 powder. Before the breakthrough experiment, the fixed bed was heated at 423 K under a flow of He for complete activation. The fixed bed was then cooled to room temperature and immersed in a water/methanol bath with different temperatures. Then, the gas mixtures ($C_2H_2/CO_2$) were introduced, and the outlet gas was monitored by mass spectrometry (Hidden QGA quantitative gas analysis system).

### In-situ synchrotron powder X-ray diffraction and structure determination
Fresh samples of Co-CUK-1 or Ni-CUK-1 were pre-activated under a dynamic vacuum at 200 °C, then loaded into a 0.7 mm borosilicate capillary under an inert atmosphere. Then the capillary was further activated by heating to 100 °C under a dynamic vacuum for 2 h before cooling down to room temperature. For samples measured under 233 K, synchrotron X-ray powder diffraction was carried out on the ID22 high-resolution powder diffraction beamline at the European Synchrotron Radiation Facility (ESRF). $C_2H_2$ or $CO_2$ was introduced into the capillary, and diffraction data were collected after one-hour stabilization. Data were measured between 0 and 35° with a 13-channel multi-analyzer stage under the wavelength of 0.354267(4) Å. Data were binned using a step size of 0.002°. For samples measured under 298 K, high-resolution powder X-ray diffraction patterns were collected on the powder diffractometer ($\lambda = 0.825829(1)$ Å) at room temperature on beamline I11 (Diamond Light Source, UK). $C_2H_2$ or $CO_2$ was dosed offline and then sealed for measurement. Data were collected between 0 and 150° using a step size of 0.001° with five multi-analyzing crystal (MAC) detectors without further re-binned.

Pawley and Rietveld refinements of the structure were carried out using the TOPAS software package (roughly between 18–0.90 Å in d-spacing). Due to the high flexibility of the framework, index with constraints was used to get the information on cell parameters and space groups. Stephen fitting was applied to describe the diffraction peaks and their anisotropic broadening. Approximate positions of the guest molecule under a rigid body model were found using the

simulated annealing approach before further refinement was used to find the optimal orientation of the guest molecules. The final refined structural parameters include cell parameters, the fractional coordinates ($x$, $y$, $z$) and isotropic displacement factors for all atoms except hydrogen, and the site occupancy factors (SOF) for guest molecules. The accuracy of the final model was verified by the convergence of the weighted profile factor ($R_{wp}$), the chemical sense of the model, and the good correlation between the observed and calculated diffraction patterns.

## Data availability
Crystallographic data for the structures reported in this article have been deposited at the Cambridge Crystallographic Data Centre under deposition numbers CCDC 2214437, 2214440–2214446. These data can be obtained free of charge via https://www.ccdc.cam.ac.uk/structures/. All the other relevant data, additional graphics, and calculations that support the findings of this study are available within the article and its Supplementary Information, or from the corresponding authors upon request.

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

## Acknowledgements

This work was supported by the Ministry of Education Singapore (MOE2019-T2-1-093, MOE-T2EP10122-0002; D.Z.), the Energy Market Authority of Singapore (EMA-EP009-SEGC-020; D.Z.), the Agency for Science, Technology and Research (U2102d2004, U2102d2012; D.Z.), the National Research Foundation (NRF-CRP26-2021RS-0002; D.Z.), EPSRC (EP/V056409/1; S.Y.), and the University of Manchester. We are grateful to Diamond Light Source and European Synchrotron Radiation Facility (ESRF) for access to Beamlines I11 and ID22, respectively. X.H. is supported by a Dame Kathleen Ollerenshaw Fellowship.

## Author contributions

D.Z. and S.Y. formulated and supervised the project. Z.Z. synthesized and characterized the materials, measured the adsorption isotherms and the breakthrough curves, and wrote the manuscript. Y.C., X.H., C.D., S.J.D., and S.Y. collected and analyzed the in-situ synchrotron X-ray diffraction data. K.C. and C.K. helped collect the dynamic breakthrough data. S.P., H.L., J.R., and X.S. contributed to the data analysis and discussion. All authors contributed to the manuscript revision.

## Competing interests

The authors declare no competing interests.
