## [Peer review file · Nature Communications]

REVIEWER COMMENTS

Reviewer #1 (Remarks to the Author):

In this work, the authors reported a series of M-CUK-1 (M = Co, Ni, and Mg) materials, which show an interesting inversion of the adsorption behavior of C₂H₂ and CO₂ under temperature control. However, the mechanism is still unclear, packing of molecules in channel is phenomenon for adsorption process, and not the reason. In my option, the work lacks the significant and novelty enough for publication in nature communications.

Reviewer #2 (Remarks to the Author):

This work reported by Zhao et al. described the tunable sorption behavior of CUK-1 for CO₂ and C₂H₂ by varying temperatures. This is for the first time revealed in porous materials. The mechanism of the results was investigated by in-situ synchrotron X-ray diffraction techniques, which clearly showed that the formation of guest clusters and guest-guest interactions would definitely play a critical role in the adsorption and separation, but was rarely reported. The adsorption and separation performances were proved by single gas adsorption and mixture gas separation experiments at different temperatures. This is a very attractive work and give new insights into the C₂H₂/CO₂ separations. It is recommended to be published in Nature Communications after the following questions to be addressed.

1. The cycling separation performance should be investigated.
2. The XRD patterns of CUK-1 after breakthrough experiments should be provided.
3. If the table listing the materials and performances for inverse CO₂/C₂H₂ separation was provided, this would be much better.
4. For readers' convenience, it is better to provide a figure of framework structures for CUK-1 in the manuscript.
5. A related paper (Angew. Chem. Int. Ed. 2021, 60, 11688, DOI: 10.1002/anie.202016673) on inverse CO₂/C₂H₂ separation should be cited and discussed.

Reviewer #3 (Remarks to the Author):

This work reports a rare normal and inverse sorption behavior of CO₂ and C₂H₂ on a series of MOFs, CUK-1, that shows normal C₂H₂/CO₂ separation under ambient conditions but inverse CO₂/C₂H₂ separation by lowering the temperature. This behavior has been fully explored for adsorption and separation at a wide temperature range. The gas clusters with different packing geometries at different temperatures in MOFs were rarely reported. Especially, the breakthrough experiments revealed that high-purity C₂H₂ could be obtained, and the materials could be well generated when the temperature increased to room temperature. This interesting behavior is good for temperature and pressure swing adsorption and separation. This work is of interest to the relevant communities and is suitable for publication in Nature Communications. The following comments should be addressed during the revision.

1. The cycling breakthrough experiments should be carried out.
2. The different sorption capacities of CO₂ and C₂H₂ at different temperatures should be listed in a table. This would provide direct information for the tunable sorption behavior of CO₂ and C₂H₂ on CUK-1.
3. The specific meaning of the different colors of dotted lines in Fig 3 and fig4 should be provided.
4. For Co-CUK-1, the breakthrough curves with the normalized y-axis (C/C₀) at 298 K

should be provided. This would provide direct information on its separation performance for C₂H₂/CO₂ mixture.

Reviewer #4 (Remarks to the Author):

Recent studies have been focused on utilization of MOFs in the carbon capture process primarily targeting CO₂ adsorption mechanism based on physisorption and chemisorption. Studies have shown that MOFs can uptake gases and these gases can be used in catalytic reaction and sorption studies. Although the use of ultra-microporous MOFs in gas capture and storage is not novel, the authors demonstrate selectivity of gases within the ultra-microporous MOFs which is a poorly explored area.

Major

The authors address the challenge of enhancing selectivity through host-guest interactions tuning. Their work reports and discusses the modulation of geometries of guest-clusters as a function of temperature for the normal and inverse selectivity and separation of CO₂ and C₂H₂ within the robust ultra-microporous M-CUK-1 (M = Co, Ni, and Mg) materials. In general, their results reveal that at low temperatures (< 253 K), CUK-1 preferentially adsorbs CO₂ with both high selectivity (> 10) and capacity (170 cm³ g⁻¹) owing to the formation of CO₂ tetramers that simultaneously maximize the guest-guest and host-guest interactions. The efficient packing of CO₂ molecules via strong guest-guest interactions forms unprecedented CO₂ clusters, leading to higher CO₂ capacity. Also, the strong host-guest interactions of CUK-1 for C₂H₂ at ambient conditions led to the preferential adsorption of C₂H₂. A temperature-dependent reversal of CO₂ and C₂H₂ adsorption affinities on CUK-1, affords high-purity C₂H₂ (99.95%) from an equimolar mixture of C₂H₂/CO₂ via a one-step purification process.

The work is novel and is significant in the field of chemical sciences. The sound methodology employed gives a flow in data analysis, interpretation and conclusion.

Minor

The authors should indicate the effect of varying M; M-CUK-1 (M = Co, Ni, and Mg) in their conclusions.

Reviewer 1

In this work, the authors reported a series of M-CUK-1 (M = Co, Ni, and Mg) materials, which show an interesting inversion of the adsorption behavior of C₂H₂ and CO₂ under temperature control. However, the mechanism is still unclear, packing of molecules in channel is phenomenon for adsorption process, and not the reason. In my opinion, the work lacks the significant and novelty enough for publication in nature communications.

Response: We thank the reviewer for the comments. Here, we would like to restate the novelty of our work, and explain the reason why CUK-1 materials exhibit tunable CO₂ and C₂H₂ sorption behavior.

To our knowledge, guest-guest interactions or guest clusters play an important role in molecular recognition systems, particularly in porous materials for gas adsorption. The exploitation of guest-guest interactions can promote the understanding of the adsorption and diffusion behavior of guest molecules in porous materials and the design of new functional porous materials. However, the prediction, control, and direct experimental observation of guest-guest interactions or guest clusters within confined nanovoids are highly challenging and remain poorly explored, especially when compared with host-guest chemistry.

Herein, we reveal the modulation of guest-cluster (CO₂ and C₂H₂) geometries as a temperature function for the tunable sorption of CO₂ and C₂H₂ within robust ultramicroporous materials. The formation of guest clusters, guest-guest interactions, and binding domains within CUK-1 material was directly visualized by in-situ synchrotron X-ray diffraction techniques. The results clearly show that the stronger host-guest interactions and the ordered arrangement of C₂H₂ clusters in the confined ultramicropores endow CUK-1 materials with the normal preferential adsorption behavior for C₂H₂ over CO₂ at room temperature. However, the one-dimensional CO₂ chains show higher pore occupancy, leading to high CO₂ capacities. The one-dimensional CO₂ chains changed to CO₂ tetramers when the temperature decreased. The ordered arrangement of CO₂ tetramers with quasi-T-shaped geometries is very similar to that of CO₂ packing in dry ice; this leads to much larger crystallographic occupancy and higher capacity increments of CO₂. In contrast, only slight changes were found for C₂H₂-cluster geometries after decreasing temperatures. The pore occupancy of C₂H₂ is always lower than that of CO₂ in CUK-1. Thus, we can see that the unique molecular arrangement or packing in the confined pore channels is the main reason for the clear sorption inversion of CO₂ over C₂H₂, and finally leads to much higher CO₂ capacities.

Furthermore, based on the DFT calculated binding energies in Supplementary Table 2 and Table 3, the binding energies of C₂H₂ at both sites are higher than those of CO₂, which is understandable for the C₂H₂ selective behavior of CUK-1. However, this cannot explain why CUK-1 shows higher CO₂ capacity at 1 bar. After the temperature decreased to 233 K, we can see that in Co-CUK-1, CO₂ binding energy at site I is higher than that of C₂H₂, but lower at site

II; in Ni-CUK-1, CO₂ binding energy at site II is higher than that of C₂H₂, but lower at site I. This observation means the binding affinity or the host-guest interaction is not the main reason for the inverse CO₂ sorption behavior at low temperatures, and this also cannot explain why the capacity for CO₂ is higher at different temperatures. Although there is a subtle difference in host-guest interactions when varying the metal nodes in CUK-1 (Supplementary Tables 2 and 3), this did not influence the formation of guest clusters or guest-guest interactions and the tunable CO₂ and C₂H₂ sorption behavior. The above results indicate that the guest-guest interactions or guest-cluster packing play a dominant role in the inverse CO₂ sorption behavior of CUK-1 materials.

Such an inverse CO₂/C₂H₂ adsorption behavior induced by the packing of molecules is desirable in the industry for C₂H₂ production via a one-step CO₂ adsorption process. The normal and inverse CO₂ and C₂H₂ sorption and separation behaviors were proved by single-component gas sorption isotherms and breakthrough experiments.

Thus, we hope the reviewer would agree that our results indicate new findings toward the rational control of the packing geometries of gas clusters in porous materials for tunable sorption and separation behavior of CO₂ and C₂H₂. For the first time, such normal and inverse sorption and separation behaviors by varying temperatures are revealed in porous materials.

Reviewer 2

This work reported by Zhao et al. described the tunable sorption behavior of CUK-1 for CO₂ and C₂H₂ by varying temperatures. This is for the first time revealed in porous materials. The mechanism of the results was investigated by in-situ synchrotron X-ray diffraction techniques, which clearly showed that the formation of guest clusters and guest-guest interactions would definitely play a critical role in the adsorption and separation, but was rarely reported. The adsorption and separation performances were proved by single gas adsorption and mixture gas separation experiments at different temperatures. This is a very attractive work and give new insights into the C₂H₂/CO₂ separations. It is recommended to be published in Nature Communications after the following questions to be addressed.

Response: We thank the reviewer for the positive comments and constructive suggestions, which have helped the improvement of our manuscript.

1. The cycling separation performance should be investigated.

Response: We thank the reviewer for the kind suggestion. The cycling breakthrough curves were collected and provided in our revised Supplementary Information as Supplementary Figure 22. The results show that CUK-1 materials exhibit excellent cycling stability for CO₂/C₂H₂ separations.

Modifications:

Supplementary Figure 22. Multiple-cycle breakthrough tests of Co-CUK-1 for CO₂/C₂H₂ (1/1) separation at 233 K.

2. The XRD patterns of CUK-1 after breakthrough experiments should be provided.

Response: We thank the reviewer for the kind suggestion. We have collected the XRD patterns of CUK-1 materials after the breakthrough experiments, as shown in Supplementary Figure 3 and Figure 4.

Modifications:

Supplementary Figure 3. PXRD patterns of Ni-CUK-1 under different conditions.

Supplementary Figure 4. PXRD patterns of Mg-CUK-1 under different conditions.

3. If the table listing the materials and performances for inverse CO₂/C₂H₂ separation was provided, this would be much better.

Response: We thank the reviewer for this suggestion. We have added a table in our revised Supplementary Information to compare the porous materials for inverse CO₂/C₂H₂ separations, as shown in Supplementary Tale 9.

Modifications:

Supplementary Table 9. Summary of the adsorption uptakes and selectivities for CO₂-selective materials.

Materials	CO ₂ uptake (cm ³ g ⁻¹)	C ₂ H ₂ uptake (cm ³ g ⁻¹)	Temperature (K)	IAST selectivity (50/50)	Ref.
Co-CUK-1	170	119	233	9.5	This work
Mg-CUK-1	144	89	233	12.1	
Cu-F-pymo	26.6	2.3	298	>10 ⁵	⁵
PMOF-1	53.3	7.5	273	694	⁶
Ce(IV)-MIL-140-4F	110.3	41.5	298	44	⁷
MUF-16	47.8	4.0	298	510	¹
CD-MOF-2	59.6	45.5	298	12.8	⁸
Tm-OH-bdc	130.6	47	298	18.2	⁹
SIFSIX-3-Ni	60.5	73.9	298	7.5	¹⁰
PCP-NH ₂ -ipa	72	43.7	298	6.4	¹¹
[Mn(bdc)(dpe)]	46.8	7.3	273	8.8	¹²

4. For readers' convenience, it is better to provide a figure of framework structures for CUK-1 in the manuscript.

Response: We thank the reviewer for this suggestion. In the part of the "configurations determined by in-situ synchrotron X-ray powder diffraction", we have provided the structures of CUK-1 materials. In order to avoid repeatedly showing the CUK-1 structures in the manuscript, we have added the original structure of CUK-1 in Supplementary Figure 1 in our revised Supplementary Information.

Modifications:

Supplementary Figure 1. (a) View of coordination mode of metal ions in CUK-1; (b) Crystal structure of desolvated CUK-1; (c) Connolly pore surface of the corrugated pore channels in CUK-1 materials.

5. A related paper (Angew. Chem. Int. Ed. 2021, 60, 11688, DOI: 10.1002/anie.202016673) on inverse CO₂/C₂H₂ separation should be cited and discussed.

Response: We thank the reviewer for this suggestion. The related reference has been added and discussed in our revised manuscript.

Modifications:

“The inversed selectivities are comparable with those of the state-of-the-art materials for inversed CO₂/C₂H₂ separation, such as [Mn(bdc)(bpe)] (9),⁴⁶ Ce(IV)-MIL-140-4F (9.6),³⁷ PCP-NH₂-ipa (6.4),³⁵ and SIFSIX-3-Ni (7.5),²⁹ but lower than the benchmark material Cu-F-pymo (> 10⁵).³⁸ Furthermore, the isosteric heat of adsorption (ΔH_{ads}) of CO₂ on Co- and Ni-CUK-1 were calculated to be 20.8 and 21.7 kJ mol⁻¹, respectively (Supplementary Fig. 10), much lower than that of other materials (Fig. 2f), such as PCP-NH₂-ipa (26.8 kJ mol⁻¹),³⁵ [Mn(bdc)(bpe)] (29.5 kJ mol⁻¹),⁴⁶ MUF-16 (32 kJ mol⁻¹),³⁹ and Tm₂(OH-bdc) (45 kJ mol⁻¹)⁴⁰.”

Reference:

“35. Gu, Y., Zheng, J., Otake, K., Shivanna, M., Sakaki, S., Yoshino, H., Ohba, M., Kawaguchi, S., Wang, Y., Li, F., *et al.* Host–guest interaction modulation in porous coordination polymers for inverse selective CO₂/C₂H₂ separation. *Angew. Chem. Int. Ed.* **60**, 11688–11694 (2021).”

Reviewer 3

This work reports a rare normal and inverse sorption behavior of CO₂ and C₂H₂ on a series of MOFs, CUK-1, that shows normal C₂H₂/CO₂ separation under ambient conditions but inverse CO₂/C₂H₂ separation by lowering the temperature. This behavior has been fully explored for adsorption and separation at a wide temperature range. The gas clusters with different packing geometries at different temperatures in MOFs were rarely reported. Especially, the breakthrough experiments revealed that high-purity C₂H₂ could be obtained, and the materials could be well generated when the temperature increased to room temperature. This interesting behavior is good for temperature and pressure swing adsorption and separation. This work is of interest to the relevant communities and is suitable for publication in Nature Communications. The following comments should be addressed during the revision.

Response: We thank this referee for these constructive comments, which, by addressing them, have helped us to improve our manuscript.

1. The cycling breakthrough experiments should be carried out.

Response: We thank the reviewer for the kind suggestion. The cycling breakthrough experiments have been carried out, and the related results were provided in our revised Supplementary Information as Supplementary Figure 22.

Modifications:

Supplementary Figure 22. Multiple-cycle breakthrough tests of Co-CUK-1 for CO₂/C₂H₂ (1/1) separation at 233 K.

2. The different sorption capacities of CO₂ and C₂H₂ at different temperatures should be listed in a table. This would provide direct information for the tunable sorption behavior of CO₂ and C₂H₂ on CUK-1.

Response: We thank the reviewer for this suggestion. We have provided a table to compare the CO₂ and C₂H₂ sorption performance of CUK-1 materials at different temperatures, as shown in Supplementary Table 1.

Modifications:

Supplementary Table 1. The sorption uptakes and packing densities of CO₂ and C₂H₂ in CUK-1 materials at 1.0 bar and different temperatures.

Temperature (K)	Pore volume (cm ³ g ⁻¹)	CO ₂ uptake (mmol g ⁻¹)		CO ₂ packing density (cm ³ g ⁻¹)		C ₂ H ₂ uptake (mmol g ⁻¹)		C ₂ H ₂ packing density (cm ³ g ⁻¹)	
		233	298	233	298	233	298	233	298
Co-CUK-1	0.24	7.59	4.7	1.39	0.86	5.20	3.82	0.56	0.41
Ni-CUK-1	0.22	6.34	3.4	1.27	0.68	4.23	3.07	0.50	0.36
Mg-CUK-1	0.23	6.45	2.7	1.24	0.52	4.02	2.70	0.45	0.31

3. The specific meaning of the different colors of dotted lines in Fig 3 and fig4 should be provided.

Response: We thank the reviewer for this suggestion. We have provided the specific meaning of the colors of the dotted lines in our revised manuscript.

4. For Co-CUK-1, the breakthrough curves with the normalized y-axis (C/C₀) at 298 K should be provided. This would provide direct information on its separation performance for C₂H₂/CO₂ mixture.

Response: We thank the reviewer for this suggestion. We have provided the normalized breakthrough curves of Co-CUK-1 materials at different temperatures, as shown in Supplementary Figures 19 and 20.

Modifications:

Supplementary Figure 19. Experimental breakthrough curves of CO₂/C₂H₂ (1/1) mixture on CUK-1 materials at 298 K.

Supplementary Figure 20. Experimental breakthrough curves of CO₂/C₂H₂ (1/1) mixture on Co-CUK-1 at 273 and 233 K.

Reviewer 4

Recent studies have been focused on utilization of MOFs in the carbon capture process primarily targeting CO₂ adsorption mechanism based on physisorption and chemisorption. Studies have shown that MOFs can uptake gases and these gases can be used in catalytic reaction and sorption studies. Although the use of ultra-microporous MOFs in gas capture and storage is not novel, the authors demonstrate selectivity of gases within the ultra-microporous MOFs which is a poorly explored area.

Major

The authors address the challenge of enhancing selectivity through host-guest interactions tuning. Their work report and discuss the modulation of geometries of guest-clusters as a function of temperature for the normal and inverse selectively and separation of CO₂ and C₂H₂ within the robust ultra-microporous M-CUK-1 (M = Co, Ni, and Mg) materials. In general, their results reveal that at low temperatures (< 253 K), CUK-1 preferentially adsorbs CO₂ with both high selectivity (> 10) and capacity (170 cm³ g⁻¹) owing to the formation of CO₂ tetramers that simultaneously maximize the guest-guest and host-guest interactions. The efficient packing of CO₂ molecules via strong guest-guest interactions forms unprecedented CO₂ clusters, leading to higher CO₂ capacity. Also, the strong host-guest interactions of CUK-1 for C₂H₂ at ambient conditions led to the preferential adsorption of C₂H₂. A temperature-dependent reversal of CO₂ and C₂H₂ adsorption affinities on CUK-1, affords high-purity C₂H₂ (99.95%) from an equimolar mixture of C₂H₂/CO₂ via a one-step purification process. The work is novel and is significant in the field of chemical sciences. The sound methodology employed gives a flow in data analysis, interpretation and conclusion.

Minor

The authors should indicate the effect of varying M; M-CUK-1 (M = Co, Ni, and Mg) in their conclusions.

Response: We thank the reviewer for this kind suggestion. After carefully analyzing the results of CUK-1 materials with different metal nodes for CO₂ and C₂H₂ sorption and separation, we found that there is only a subtle difference in host-guest interactions between CO₂ or C₂H₂ and CUK-1 materials when varying the metal nodes (Supplementary Tables 2 and 3), which would not influence the formation of guest clusters (Figures 3 and 4 in our manuscript, Supplementary Figures 15 and 16). This means the guest-guest interactions or guest-cluster packing play the dominant role in the tunable inverse CO₂ sorption behavior of CUK-1 materials. The related discussion has been added to our revised manuscript.

Modifications:

“There is a subtle difference in host-guest interactions when varying the metal nodes in CUK-1 (Supplementary Tables 2 and 3), but this did not influence the formation of guest clusters and the tunable CO₂ and C₂H₂ sorption behavior. This means the guest-guest interactions or guest-cluster packing play the dominant role in the inverse CO₂ sorption behavior of CUK-1 materials.”

REVIEWERS' COMMENTS

Reviewer #1 (Remarks to the Author):

The authors has fully revised the manuscript and this paper can published in Nature Communications.

Reviewer #2 (Remarks to the Author):

The authors have well addressed the concerns. Now this paper can be published.

Reviewer #3 (Remarks to the Author):

The authors have addressed the reviewers' comments and the revised version is suitable for publication.

Reviewer #4 (Remarks to the Author):

The revised version of the manuscript addresses my comments and concerns fully. I therefore, recommend the editors to consider publishing the manuscript.